# Association between HLA Class II Alleles/Haplotypes and Genomic Ancestry in Brazilian Patients with Type 1 Diabetes: A Nationwide Exploratory Study

**DOI:** 10.3390/genes14050991

**Published:** 2023-04-27

**Authors:** Marília Brito Gomes, Vandilson Rodrigues, Deborah Conte Santos, Paulo Ricardo Villas Bôas, Dayse A. Silva, Rossana Santiago de Sousa Azulay, Sergio Atala Dib, Elizabeth João Pavin, Virgínia Oliveira Fernandes, Renan Magalhães Montenegro Junior, João Soares Felicio, Rosangela Réa, Carlos Antonio Negrato, Luís Cristóvão Porto

**Affiliations:** 1Department of Internal Medicine, Diabetes Unit, Rio de Janeiro State University (UERJ), Rio de Janeiro 20950-003, Brazil; 2Research Group in Clinical and Molecular Endocrinology and Metabology (ENDOCLIM), São Luís 65080-805, Brazil; 3Histocompatibility and Cryopreservation Laboratory (HLA), Rio de Janeiro State University (UERJ), Rio de Janeiro 20950-003, Brazil; 4DNA Diagnostic Laboratory (LDD), Rio de Janeiro State University (UERJ), Rio de Janeiro 20550-900, Brazil; 5Service of Endocrinology, University Hospital of the Federal University of Maranhão (HUUFMA/EBSERH), São Luís 65020-070, Brazil; 6Endocrinology Division, Escola Paulista de Medicina, Federal University of São Paulo (UNIFESP), São Paulo 04023-062, Brazil; 7Endocrinology Division, School of Medical Sciences, University of Campinas (UNICAMP), São Paulo 13083-970, Brazil; 8Department of Clinical Medicine, Federal University of Ceará (UFC), Fortaleza 60430-275, Brazil; 9Department of Community Health, Federal University of Ceará (UFC), Fortaleza 60430-275, Brazil; 10Clinical Research Unit, Walter Cantídio University Hospital, Federal University of Ceará (UFC/EBSERH), Fortaleza 60430-372, Brazil; 11Endocrinology Division, João de Barros Barreto University Hospital, Federal University of Pará (UFPA), Belém 66073-000, Brazil; 12Endocrinology Unit, Federal University of Paraná (UFPR), Curitiba 80060-900, Brazil; 13Medical Doctor Program, School of Dentistry, University of São Paulo (USP), Bauru 17012-901, Brazil

**Keywords:** human leukocyte antigen, HLA, type 1 diabetes, ethnicity, genomic ancestry, admixed

## Abstract

We aimed to identify *HLA-DRB1*, *-DQA1*, and *-DQB1* alleles/haplotypes associated with European, African, or Native American genomic ancestry (GA) in admixed Brazilian patients with type 1 diabetes (T1D). This exploratory nationwide study enrolled 1599 participants. GA percentage was inferred using a panel of 46 ancestry informative marker-insertion/deletion. Receiver operating characteristic curve analysis (ROC) was applied to identify HLA class II alleles related to European, African, or Native American GA, and showed significant (*p* < 0.05) accuracy for identifying HLA risk alleles related to European GA: for *DRB1*03:01*, the area under the curve was (AUC) 0.533; for *DRB1*04:01* AUC = 0.558, for *DRB1*04:02* AUC = 0.545. A better accuracy for identifying African GA was observed for the risk allele *DRB1*09:01*AUC = 0.679 and for the protective alleles *DRB1*03:02* AUC = 0.649, *DRB1*11:02* AUC = 0.636, and *DRB1*15:03* AUC = 0.690. Higher percentage of European GA was observed in patients with risk haplotypes (*p* < 0.05). African GA percentage was higher in patients with protective haplotypes (*p* < 0.05). Risk alleles and haplotypes were related to European GA and protective alleles/haplotypes to African GA. Future studies with other ancestry markers are warranted to fill the gap in knowledge regarding the genetic origin of T1D in highly admixed populations such as that found in Brazil.

## 1. Introduction

Type 1 diabetes (T1D) is the commonest chronic endocrine disorder observed in young people caused by autoimmune destruction of pancreatic β-cells [1,2], and it has an ongoing increase in incidence in developed and developing countries [3]. In Brazil, a country composed of a highly admixed population of 209 million people, the estimated incidence according to the International Diabetes Federation (IDF) was 7.6/100,000 persons/year [3]. However, in some Brazilian cities, such as Bauru (São Paulo State), epidemiological findings showed a higher incidence that reached almost 12.8/100,000 [4]. So far, the main hypothesis for T1D pathogenesis is that the disease is caused by an interaction between multiple genes and environmental factors [2]. Concerning genetics, the alleles from the HLA region account for almost 50% of the genetic risk, especially class II alleles, DR, and DQ, even though class I alleles and non-HLA alleles also increase this risk [5].

The susceptibility to T1D is mainly linked with the haplotypes *DRB1*03:01~DQA1*05:01~DQB1*02:01* and *DRB1*04~DQA1*03:0~DQB1*03:02,* although the odds risks vary significantly among different populations [6,7]. Certain HLA haplotypes, such as *DRB1*07:01~DQA1*03:01~DQB1*02:02*, may increase the risk of T1D in African-Americans [8]. However, some changes in the *HLA-DQA1* allele, such as the *DQA1*02:01* allele, which is known to be protective for the European population [2,5], is one of the most common alleles found in the general Brazilian population [9]. Until now, the majority of T1D studies have been conducted on homogeneous populations with low genetic variability, specifically Caucasians with European ancestry [1,2,3,4,5,6]. These studies have often relied on self-reported ethnicity rather than genetic ancestry to categorize patients according to ethnicity [1,2,3,4,5,6]. However, this approach can introduce biases in allele frequency studies, especially in admixed populations [10,11,12].

The above-mentioned evidence must be evaluated in the context of the rise of genetic risk scores for diagnosing and predicting of T1D in different populations [13,14,15]. However, due to ethnic differences among populations, the studies that take into account the genomic ancestry (GA) of T1D could add more information to better individualize these genetic risk scores, mainly in admixed populations. Brazil is one of the countries with the highest genetic variability due to hundreds of years of miscegenation [16]. Before the colonization period, which started in 1500, Brazil was composed of Native Americans. With the arrival of Portuguese and other European populations, miscegenation started. Some years later, with the beginning of the slavery traffic, the African population also started to contribute to the substantial Brazilian genetic variability [11,17].

A previous study of our group that compared the GA of patients with T1D with the general Brazilian population, showed a predominance of European ancestry in both groups, especially in patients with T1D [11]. In that study, we found a low proportion of Native American ancestry and only 6.2% of the studied people presented ratios greater than 95% for one single ancestry. Moreover, the European GA was the only one that had a high proportion (>95%) in that sample [11]. Polymorphisms of classical alleles of the HLA system could vary among populations from different regions of a continental country such as Brazil due to the great variability of the GA contribution caused by the different migration waves and the frequency of Native American ancestry [8,18]. Therefore, the study of HLA genes and their relation with GA can add important information for understanding the origins of populations and their degree of admixture [19] and could help understanding the characteristics of T1D in our country.

In this study, we hypothesized that different HLA class alleles can be related to the European, African, or Native American ancestry percentage in Brazilian people with T1D. In this sense, this study aimed to identify which risk or protective *HLA-DRB1*, *-DQA1*, and *-DQB1* alleles/haplotypes were associated with European, African, or Native American ancestry percentage in a sample of Brazilian patients with T1D.

## 2. Methods

### 2.1. Study Design and Sample

The present exploratory study was conducted with 1599 unrelated patients with T1D who participated in a nationwide multicenter cross-sectional study, which originally enrolled 1912 patients and was performed between August 2011 and August 2014 in 14 public clinics, located in 11 Brazilian cities from five geographical regions (North, Northeast, Midwest, Southeast, and South). The flowchart of the study is described in Figure 1. The methodology was previously described [20]. Briefly, all patients T1D received health care from the Brazilian National Health Care System (SUS) and were diagnosed based on typical clinical signals and symptoms of T1D, which are hyperglycemia, weight loss, polyuria, polydipsia, polyphagia, and the requirement of continuous insulin use ever since [20]. Patients who were at least 13 years old and followed up at their respective diabetes center for a minimum of 6 months were included. Each clinic provided data from at least 50 T1D outpatients who regularly attended the clinic.

This study was approved by the ethics committee of Pedro Ernesto University Hospital (Rio de Janeiro State University) and by the local ethics committee of each center. Written informed consent was obtained from all patients or their parents where necessary.

### 2.2. Data Collection

In a clinical appointment, through the application of a standardized questionnaire, clinical and demographic information were collected, including gender, current age, birthplace, self-reported color-race, age at diagnosis, and diabetes duration. Color-race was assessed using the classification system of the Brazilian Institute of Geography and Statistics (IBGE), which categorizes individuals as Black (preta), White (branca), Brown (parda), Yellow (of Asian origin) (amarela), or Indigenous (Native American origin) (indígena) [21].

### 2.3. DNA Extraction and AIM-Indel Genotyping

Peripheral blood genomic DNA was extracted using the SP QIA Symphony commercial kit and QIA Symphony equipment, following the manufacturer’s instructions (Qiagen, Germantown, MD, USA). A panel of 46 AIM-INDEL was used to infer global and individual GA, according to a previously described protocol [22]. These markers present varying allele frequencies of European, African, and Native American populations. Genotyping was carried out using multiplex PCR followed by capillary electrophoresis with the ABI 3500 system, and allele naming was performed using Gene Mapper V.4.1 software (Applied Biosystems by Life Technologies, Carlsbad, CA, USA). Ancestry was estimated using Structure V.2.3.3 software, with the HGDP-CEPH diversity panel (Sub-Set H952) as a reference for ancestral populations.

To compare allele frequency of the genotyped 46 AIM Indels and the ancestry percentage of patients with T1D, we used published data from the Brazilian population for the same markers. This dataset contained 936 unrelated healthy individuals from different metropolitan areas [17], and the comparison was performed as described previously [8].

### 2.4. HLA Genotyping

HLA class II alleles (*HLA-DRB1*, *DQA1*, and *DQB1*) were genotyped according to the method described by Santos et al. [7]. DNA was extracted from participants, and 352 samples were amplified at loci *HLA-DRB1* and *-DQB1* using primers from the NGSgo^®^ v2 library preparation kit (GenDx, Utrecht, The Netherlands). An additional 124 samples were genotyped using the Holotype HLA Assay (Omixon Inc., Budapest, Hungary) for *HLA-DRB1*, *-DQB1*, and *-DQA1*, following the manufacturer’s instructions. The remaining 1123 samples were typed using Medium to High-resolution PCR-reverse sequence specific oligonucleotide (PCR-RSSO) (LabType SSO, One lambda Inc., West Hills, CA, USA). Alleles were defined using Common, intermediate, and well-documented (CIWD) frequencies [23], and sequencing was used to resolve any ambiguities. Three *locus* haplotype frequencies (*DRB1~DQA1~DQB1*) were estimated for each self-reported color-race and region, resolving phase and allelic ambiguity using the expectation-maximization (EM) algorithm. Deviations from Hardy–Weinberg equilibrium (HWE) were assessed at the allele-family level (first nomenclature field) using a modified version of the Guo and Thompson algorithm as implemented in the software Arlequin v.3.5 [24]. *HLA-DQA1* alleles were imputed in 31.5% of the samples from the T1D group (*n* = 321) using the linkage disequilibrium criteria, based on the results found by NGS. After validating the HLA dataset via an EM algorithm for resolving allelic ambiguities and determining both allele and extended haplotype frequencies, this imputation was manually performed according to the haplotype results from Arlequin output data, stratified by self-reported color-race and regions. Due to a large number of haplotypes found, only those with a total count of ten or greater were presented in the table. Rare haplotypes were labeled as “other”.

### 2.5. Statistical Analysis

Data were analyzed using the Stata software version 18 (Stata Corp., College Station, TX, USA) and GraphPad Prism software version 9 (GraphPad Software Inc., San Diego, CA, USA). Descriptive statistics were expressed as frequency and percentage for categorical data, whereas continuous data were expressed as mean, median, standard deviation (±SD), interquartile interval, and range (minimum-maximum).

Receiver operating characteristic curve (ROC) analysis was applied to identify HLA class II alleles related to an increased percentage of European, African, or Native American ancestry and to estimate the area under the ROC curve (AUC) and 95% confidence interval. Additionally, the percentage threshold of European, African, or Native American ancestry related to each HLA class II allele were those with the optimal Youden index [25], determined as the ancestry percentage at which the sum of sensitivity and specificity was maximal. Comparative analysis of global GA percentage between genotype groups was performed using One-way ANOVA followed by Tukey’s multiple comparisons test. Bar charts, box-plots, and tables were used to present the analyzed data. The risk or protective haplotypes groups were categorized based on data from a previous study that investigated haplotypes associated with risk or protection for T1D in the Brazilian population [7] (Appendix A). DQ2 corresponding to *DQA1*05:01~DQB1*02:01* and DQ8 to *DQA1*03:01~DQB1*03:02*. The significance level was 5% (*p* < 0.05).

## 3. Results

### 3.1. Description of the Study Sample

A total of 1599 patients with T1D (46.2% males and 53.8% females) with a mean age of 29.7 ± 11.9 years were included in the study. As shown in Table 1, most of the patients in the sample were White (51.5%) and Brown (38.5%), according to the self-reported skin-color/race.

### 3.2. Global Genomic Ancestry Percentage and HLA Allele Distribution

In this analysis, there was a higher average percentage of European ancestry (64.2 ± 21.6, ranging from 2.3% to 99.2%), followed by African (21.0 ± 17.4, ranging from 0.2 to 88.7) and Native American (14.8 ± 11.9, ranging from 0.4 to 69.5) (Figure 2a,b). Distribution of ordinal categories showed that 74.8% of patients had more than 50% of European ancestry, while only 7.1% and 0.8% had more than 50% African and Native American ancestry, respectively (Figure 2c).

Figure 3 shows the frequency distribution of alleles in the sample. The four most frequent *HLA-DRB1* alleles were *03:01* (29%), *04:05* (11%), and *07:01* (7.5%). The most frequent *HLA-DQA1* alleles were *03:01* (35%), *05:01* (34.3%), and *01:01* (9.4%). In addition, the most frequent *HLA-DQB1* alleles were *02:01* (28.9%), *03:02* (28.5%), and *02:02* (10.9%).

### 3.3. HLA Alleles Related to Global Ancestry Percentage

The ROC curve analysis was performed, and the details of the estimated parameters are shown in Appendix A while the main findings are highlighted in Figure 4. Statistical analysis showed that an increase in the European ancestry percentage was associated with the alleles: *DRB1*03:01* (AUC = 0.533, *p* = 0.021, threshold >58.4%), *DRB1*04:01* (AUC = 0.558, *p* = 0.009, threshold > 91.6%), *DRB1*04:02* (AUC = 0.545, *p* = 0.037, threshold ≥ 46.3%), *DRB1*08:01* (AUC = 0.583, *p* = 0.032, threshold ≥ 50.8%), *DQA1*05:01* (AUC = 0.533, *p* = 0.022, threshold > 60.9%), *DQB1*02:01* (AUC = 0.536, *p* = 0.013, threshold > 58.2%), *DQB1*03:02* (AUC = 0.529, *p* = 0.045, threshold > 44.9%).

An increased percentage of African ancestry was related to *DRB1*03:02* (AUC = 0.649, *p* = 0.016, threshold > 48.3%), *DRB1*08:04* (AUC = 0.614, *p* < 0.042, threshold >16.5%), *DRB1*09:01* (AUC = 0.679, *p* < 0.001, threshold > 26.1%), *DRB1*11:02* (AUC = 0.636, *p* = 0.048, threshold > 35.9%), *DRB1*15:03* (AUC = 0.690, *p* < 0.001< threshold > 13.4%), *DQA1*01:02* (AUC = 0.551, *p* = 0.012, threshold > 37.8%), *DQB1*02:02* (AUC = 0.543, *p* = 0.015, threshold > 30.7%), *DQB1*03:01* (AUC = 0.547, *p* = 0.031, threshold > 35.3%), *DQB1*06:02* (AUC = 0.594, *p* = 0.015, threshold > 37.8%) (Appendix A).

Native American ancestry increased percentage was related to *DRB1*04:04* (AUC = 0.552, *p* = 0.047, threshold > 7.9%), *DRB1*04:07* (AUC = 0.652, *p* = 0.022, threshold > 6.5%), *DRB1*08:02* (AUC = 0.648, *p* = 0.048, threshold > 23.4%), *DQB1*05:07* (AUC = 0.702, *p* = 0.021, threshold > 23%). In addition, *DRB1*16:02* was related to both African (AUC = 0.678, *p* = 0.003, threshold > 15.4%) and Native American (AUC = 0.621, *p* = 0.046, threshold >26.2%) increased percentages (Appendix A). Furthermore, Appendix A show the descriptive statistics of global ancestry percentage according to HLA class II alleles.

### 3.4. HLA Haplotypes Associated with T1D and Global Genomic Ancestry Percentage

Figure 5 illustrates the frequency of risk and protective haplotypes for T1D in the sample. The three most frequent risk haplotypes were *03:01~05:01g~02:01* (46.7%), *04:05~03:01g~03:02* (19.8%), and *04:02~03:01g~03:02* (12.4%) (Figure 5a). The three most frequent protective haplotypes were *07:01~02:01~02:02* (12.4%), *01:02~01:01~05:01* (5.6%) and *13:01~01:03~06:03* (4%) (Figure 5b). In the total sample, the risk genotype frequency was of 61.7%. In the comparative analysis, there was a higher European ancestry percentage in the risk groups than in the protective groups (*p* < 0.05). On the other hand, the African ancestry percentage was higher in the protective group than in the risk group (*p* < 0.05). The Native American ancestry percentage had no statistical difference between groups.

Appendix A shows the descriptive statistics of global ancestry percentage according to HLA haplotypes. Considering the risk haplotypes, the lowest mean of European ancestry was found in the risk haplotype *09:01~03:01~02:02* in comparison to DRB1**03:01*~DQ2 and *DRB1*04:01/02/04/05*~DQ8. The former haplotype had the highest mean of African ancestry.

Considering protective haplotypes, the lowest mean of European ancestry was found in the following haplotypes: *03:02~04:01~04:02*; *11:01~01:02~06:02,* and *15:03~01:02~06:02*; *08:02~04:01~04:02*. These haplotypes have the highest mean of African and Native American ancestry.

Overall, the highest mean of European ancestry was found in *16:01~01:02~05:02*, *04:02~03:01~03:02*, *13:01~01:03~06*:03, and *04:01~03:01~03:02*. The highest mean of African ancestry was found in *11:01~01:02~06:02* and *09:01~03:01~02:02*. The highest mean of Native American ancestry was found in *03:02~04:01~04:02* and *08:02~04:01~04:02.*

## 4. Discussion

The present study seems to be the first one carried out that analyzed the relationship between GA and alleles/haplotypes/genotypes of the HLA system while using Brazilian patients having T1D who belong to one of the most highly admixed populations worldwide. The study findings showed that most of the studied patients (~75%) had over 50% of European GA contrasting with the lower percentage of patients that had over 50% of African GA and Native American GA, 7% and 1%, respectively. Moreover, the most important risk alleles, *DRB1*03:01* and *DRB1*04:01/02*, had also a higher percentage of European GA. Moreover, patients with risk HLA haplotypes and with the combination of risk/protective haplotypes for T1D showed higher European GA contrasting with the higher African GA observed in patients with protective HLA haplotypes. These findings highlight that the increase in European GA can be associated with HLA alleles and risk haplotypes for Brazilian patients with T1D.

The data of the present study must be viewed in the context of the history of Brazil, a country with over 200 million people with a great genetic diversity due to an interethnic admixture among three main ethnic roots: European, African, and Native American for more than five centuries. However, it is important to emphasize that Brazil was originally inhabited only by Native Americans before its discovery by the Portuguese in 1500. Afterwards, a great Portuguese-Native American admixture started, followed by Portuguese-African that started with the arrival of African slaves in the 16th century, and finally with Europeans that came mostly as immigrants in the 17th and 18th centuries [19]. The latter event was related to a political process known as the “whitening of Brazilian population” (1872–1975), when approximately 5 million immigrants from different parts of Europe came to live in our country in about 100 years [19,21]. Although the majority of these immigrants were Italians and Iberians, an important percentage of them were also from other regions of Europe (Central and Eastern).

The aforementioned data must be aligned with the actual propose for the rule of HLA system, a system with a high level of polymorphism and genetic variation worldwide, which could be linked with significant signals of human geographic expansion, demographic history, and cultural diversification. Our results are in agreement with Sanchez-Mazas’ group statement in the complementary information of the HLA system as a tool for anthropological studies [26]. In this context, the Brazilian population, which was predominantly Black and Brown until 1872, have received different Caucasian HLA alleles from Europe in a short period of time [27]. Our results showed a great diversity of HLA alleles distribution, most of them having already been described in Europeans [28], Africans [29], and Native Americans [30], which is in accordance with our three main ethnic roots [16]. Some risk or protective HLA alleles were associated with an increased percentage of European, African, and Native American GA. In addition, we have observed a higher percentage of European GA in patients carrying risk haplotypes. It is noteworthy the higher percentage of African GA found in patients carrying protective haplotypes. Considering the overall studied population, 77.7% had some risk HLA haplotype that had been already described in Caucasians patients with T1D from Europe, USA, India, South America, Africa, and Asia [31].

Interestingly, the frequency of *DRB1*09:01* was low in the study sample (4.8%), and its detection was related to increased African ancestry. These findings can be supported by previous studies that have identified an association between the *DRB1*09:01* allele and T1D in African-Americans [9] and in Africans from Mali [29]. In general, this allele conferred risk in the presence of Caucasian risk alleles in the genotypes, such as *DRB1:03:01* and *DRB1*04* [7,9]. Although considered as an African allele, this allele was also observed in Native Americans (from Brazil, Paraguay, Bolivia, and Argentina) [32,33], Malaysian Indigenous [34], New Zealand Maori [35], and general population of West Russia (Siberia) [36]. The haplotype *09:01~03:01~02:02* had the highest mean of African ancestry among the risk haplotypes evaluated in this study. Three other HLA alleles, *DRB1*03:02*, *DRB1*15:01*, and *DRB1*15:03*, were among the ten most frequent protective alleles in the present study and were also described as protective alleles in African-Americans with T1D [9]. These latter HLA-DRB1 alleles had the highest African GA in our sample.

The ROC curve analysis, although significant, showed low accuracy for identifying risk HLA alleles for T1D related to increased percentage of European ancestry (*DRB1*03:01*, *DRB1*04:01*) and better accuracy for identifying risk allele *DRB1:09:01* and the protective alleles and haplotypes *DRB1*03:02*, *11:01~01:02~06:02*, and *15:03~01:02~06:02*; *08:02~04:01~04:02*, all related to an increased African ancestry in Brazilian patients with T1D. The findings observed with the ROC curve analysis could be probably related to the great admixture of the Brazilian population, mainly between White and other ethnic groups, such as Black and Native Americans, which could have resulted in a great diversity of allelic groups. For instance, the allele *DRB1*16:02* could be an example: the ROC curve analysis showed similar accuracy for African and Native American GA, and this allele had one of the highest percentages of African GA (32.3%) and Native American (19.5%), in our database. This fact probably resulted from the admixture between Native Americans and Blacks during the slave trade that occurred in our country between the 17th and 19th centuries [37]. This allele was described in the general population of West Russia, Siberia [36], New Zealand Maori [35], and Native Americans from the USA, Mexico, Brazil, Bolivia, Argentina, Paraguay [32,33,34,35], and African-Americans [9]. So far, it had not been described in the African general population.

It is beyond the scope of the present study to formulate a hypothesis about genetic penetrance of risk alleles of the HLA system in patients with T1D, but it is noteworthy that some patients carrying the risk alleles of the HLA system had a lower percentage of European GA (Appendix A).

In the present study, we did not formulate a hypothesis regarding genetic penetrance of risk alleles of the HLA system in patients with T1D. However, it is noteworthy that some patients carrying risk alleles of the HLA system had lower percentage of European GA (Appendix A). In a previous study, we found an increased odds of presenting the allele *DRB1*03:01* only in those patients who reported having all White relatives [38].

The current study has several strengths. It is the first multicenter study that included a large multi-ethnic sample of patients with T1D from all geographical regions of Brazil, characterized by a great genetic diversity and racial admixture. We used a uniform, standardized recruitment protocol in all participating centers and performed the genotyping of the three loci of the HLA system: HLA-DRB1, -DQA1, and -DQB1 in all studied patients. However, this study also has some limitations that need to be mentioned. We used only clinical criteria to define T1D, and we did not measure autoantibodies against pancreatic islet β cells or serum C peptide levels. This could have led to the misclassification of some patients as having T1D. Nonetheless, the use of clinical criteria to define T1D is common in epidemiologic studies such as ours [6,9].

Therefore, future studies analyzing other genetic markers associated with T1D could enhance the accuracy of ROC analysis and help to screen more genetic profiles associated with susceptibility to T1D.

## 5. Conclusions

The present study showed that in Brazilian patients with T1D, the most frequent alleles/haplotypes were those that have already been described as risk or protective alleles in multicenter studies conducted worldwide, mainly in Caucasian populations.

Moreover, the risk alleles and haplotypes were related to an increased percentage of European ancestry contrasting with the higher percentage of African ancestry observed in protective alleles/haplotypes.

However, future studies with other ancestry markers are needed to fill the knowledge gap of the genetic origin of T1D in admixed populations such as the Brazilian.

## Figures and Tables

**Figure 1 genes-14-00991-f001:**
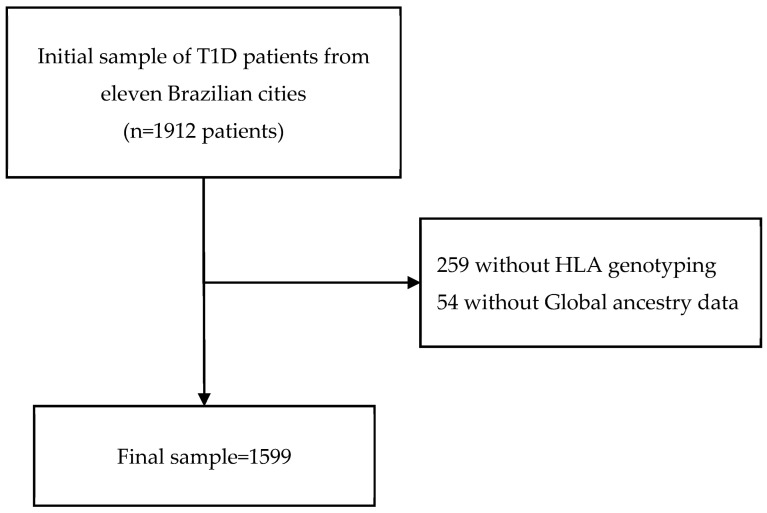
Flowchart of studied patients with type 1 diabetes.

**Figure 2 genes-14-00991-f002:**
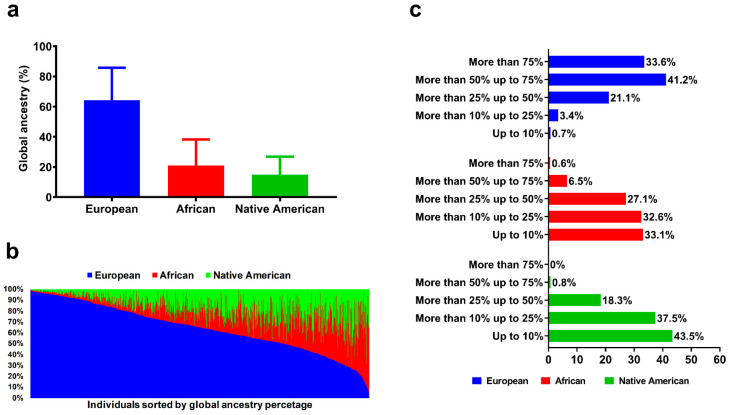
Descriptive analysis of global genomic ancestry percentage in a Brazilian sample of patients with type 1 diabetes. (**a**) Mean and standard deviation of European, African, and Native American ancestry percentage. (**b**) Individual global ancestry sorted in ascending order of European ancestry percentage. (**c**) Distribution of European, African, and Native American ancestry percentages sorted in different cutoff categories.

**Figure 3 genes-14-00991-f003:**
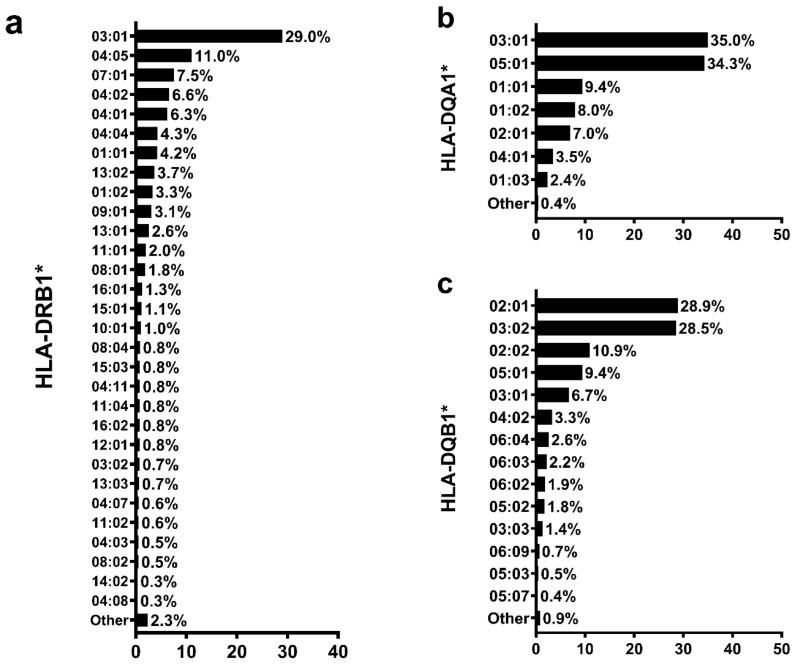
Distribution of allele frequency of HLA class II in Brazilian patients with type 1 diabetes (2n = 3198). (**a**) Allele frequency of *HLA-DRB1*. (**b**) Allele frequency of *HLA-DQA1*. (**c**) Allele frequency of *HLA-DQB1*.

**Figure 4 genes-14-00991-f004:**
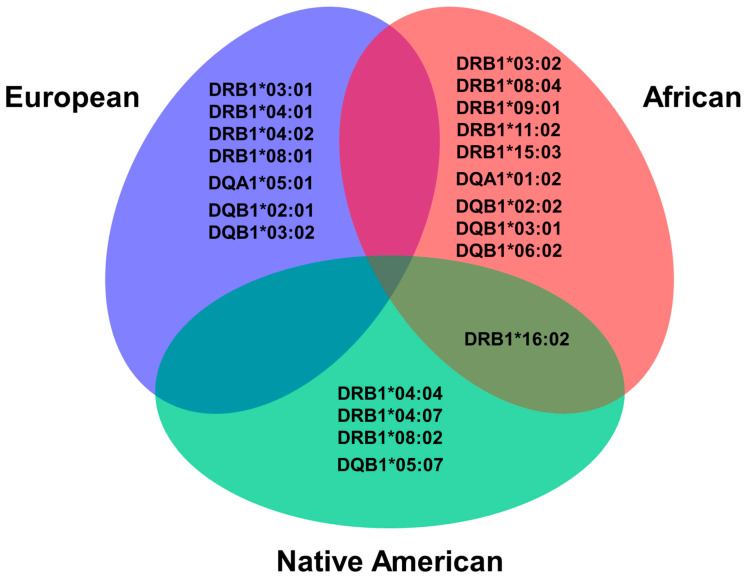
HLA class II alleles associated with increased percentage of European, African, or Native American genomic ancestry in the Brazilian patients with type 1 diabetes (*p* < 0.05 in AUC analysis).

**Figure 5 genes-14-00991-f005:**
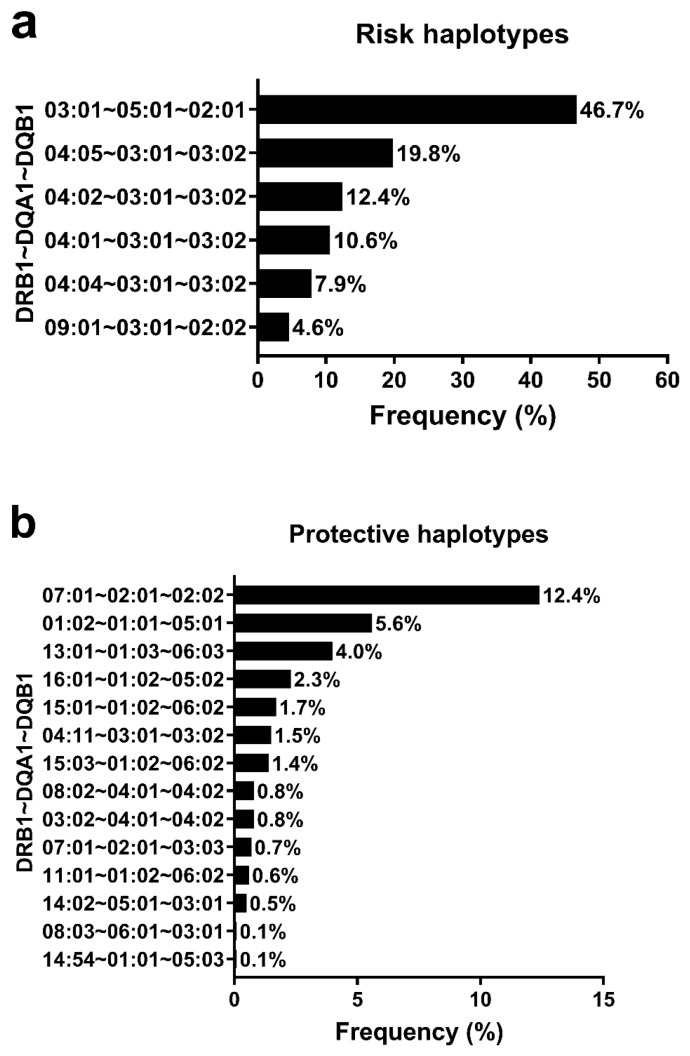
Distribution of risk (**a**) and protective (**b**) haplotypes for patients with type 1 diabetes (*n* = 1599).

**Table 1 genes-14-00991-t001:** Distribution of demographics and diabetes data (*n* = 1599).

Variables	*n*	(%)
Sex		
Male	738	(46.2)
Female	861	(53.8)
Age		
<20 years old	360	(22.5)
≥20 years old	1239	(77.5)
Age at type 1 diabetes diagnosis (mean ± SD)	14.8 ± 8.9	
Age group at type 1 diabetes onset		
0–4 years	145	(9.1)
5–9 years	328	(20.5)
10–14 years	442	(27.6)
15–19 years	281	(17.6)
20–24 years	158	(9.9)
25–29 years	134	(8.4)
≥30 years	111	(6.9)
Brazilian region of birth		
Southeast	680	(42.5)
Northeast	539	(33.7)
South	208	(13.0)
Midwest	127	(7.9)
North	45	(2.8)
Self-reported skin-color/race		
White	824	(51.5)
Brown	616	(38.5)
Black	130	(8.1)
Yellow	16	(1.0)
Indigenous	13	(0.8)

Data are presented as mean, standard deviation, or frequency, *n* (%) or mean and standard deviation (±SD).

## Data Availability

The datasets generated during and/or analyzed during the current study are available with the corresponding author upon reasonable request.

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
