# Peer review of "Association between HLA Class II Alleles/Haplotypes and Genomic Ancestry in Brazilian Patients with Type 1 Diabetes: A Nationwide Exploratory Study"

_genes, 2023, doi:10.3390/genes14050991_

Round 1

Reviewer 1 Report

In this manuscript, the authors enrolled 1,599 participants of T1D and analyzed the characteristics and relationship of HLA class II alleles and genomic ancestry. The authors found that HLA risk or protective alleles were significantly related to European GA or African GA. It’s of great interest to see the cross-sectional study of the associations of HLA alleles with genomic ancestry in admixed population with T1D. Please see the comments below:

1. It will be more comprehensive to include data of type 2 diabetes and control subjects. The characteristics of HLA in non-T1D population in Brazil are not known and it’s hard to conclude the relationship between HLA and GA in T1D. The study didn’t have islet autoantibody information, so it’s possible to mix some participants with T2D, and a group of T2D should be set up to compare with T1D subjects.

2.  It’s better to analyze subgroups with different ages of T1D-onset.  For 13 years older was one of the inclusion criteria, it’s possible to mix some LADA patients, and LADA has different HLA risk alleles.

Author Response

Response to Reviewer 1 Comments

RE: genes-2341716

Thank you for the opportunity to revise our manuscript titled “Association between HLA class II alleles/haplotypes and genomic ancestry in Brazilian patients with type 1 diabetes: a nationwide exploratory study”.

We have provided a point-by-point response to the reviewers’ comments and have amended our manuscript accordingly. Changes in the original version are tracked in the manuscript and marked in red in the point-by-point response below.

We hope that the revised manuscript addresses adequately all reviewers’ comments and is now suitable for publication in Genes. If the reviewers have any further comments, we would be happy to consider them. We look forward to hearing from you.   

Yours sincerely,

Reviewer #1: In this manuscript, the authors enrolled 1,599 participants of T1D and analyzed the characteristics and relationship of HLA class II alleles and genomic ancestry. The authors found that HLA risk or protective alleles were significantly related to European GA or African GA. It’s of great interest to see the cross-sectional study of the associations of HLA alleles with genomic ancestry in admixed population with T1D. Please see the comments below:

Point 1: It will be more comprehensive to include data of type 2 diabetes and control subjects. The characteristics of HLA in non-T1D population in Brazil are not known and it’s hard to conclude the relationship between HLA and GA in T1D. The study didn’t have islet autoantibody information, so it’s possible to mix some participants with T2D, and a group of T2D should be set up to compare with T1D subjects.

Response 1: We thank the reviewer for your careful reading of the manuscript. We agree but unfortunately, our sample comprises only patients with type 1 diabetes. We have already put this fact as a limitation of our study as follows: “We used only clinical criteria to define T1D. We did not measure autoantibodies against pancreatic islet beta cells or serum C peptide levels, which could have led us to misclassify some patients as having T1D. However, the use of clinical criteria to define T1D is common in epidemiologic studies like ours [6][9].”

Point 2: It’s better to analyze subgroups with different ages of T1D-onset.  For 13 years older was one of the inclusion criteria, it’s possible to mix, and LADA has different HLA risk alleles.

Response 2: We agree with the reviewer's comments. However, as above-mentioned we used only clinical criteria to define T1D. We have included the distribution of the age group at T1D onset (Table 1) to clarify this point. Age at onset does not comprise our current objectives. It will be interesting to study age at T1D onset in a future investigation.

Best regards, the Authors.

Reviewer 2 Report

The manuscript ”Association between HLA class II alleles/haplotypes and genomic ancestry (GA) in Brazilian patients with type 1 diabetes: a nationwide exploratory study” by Gomes and co-workers describes a large nationwide study in Brazil where they aimed to identify HLA-DRB1, .-DQA1 and -DQB1 alleles/haplotypes associated with European, African and Native American genomic ancestry in Brazilian patients with type 1 diabetes (T1D). They used a panel of 46 insertion/deletion markers around human genome developed to identify genomic ancestry of studied subjects. This analysis was able to measure the percentage of each genomic ancestry in individual genomes and revealed the strong admixture in the population. The study revealed that as a rule a higher percentage of European GA was observed in patients with risk alleles and haplotypes whereas African GA was higher in patients with protective alleles and haplotypes. The observed differences in risk and protective alleles and haplotypes are in concordance with earlier studies in different genetic backgrounds but the new approach able to genetically define the percentage of different genetic ancestries at individual level may be useful in estimation of T1D risk in admixed populations.

Minor points:

Introduction, page 3, row 68: …”the classes I alleles”…. Should be: the class I alleles.

Rows 74 and 76: Citations 8 and 9 in text are apparently changed in the reference list.

Rows 85 and 86: …”take in account”….Should be “take into account”

Page 4, row 98:…”Moreover, this latter event was…”.   This might be rephrased

Methods, HLA genotyping, page 6, row 179: …”PCR-RSSO”…. RSSO should be opened

Row 180:….CWID….. CWID should be opened.

Page 9, row 228…”In this sample”…. The word sample might be replaced by e.g. analysis.

Discussion, page 15, rows 340-341: “The aforementioned data must be aligned with the actual propose for the rule of HLA system”,      This sentence should be clarified.

Rows 359-360: “In general, the risk of this allele is related to the presence of Caucasian alleles in the haplotype (DRB1:03:01 and DRB1*04)”…This is unclear. How can the risk of DRB1*09:01 be related to the presence of DRB1*03:01 and DRB1*04?

Page 16, row 372:  “and the protective alleles”… There is one protective allele and then haplotypes. Should apparently be “protective alleles and haplotypes”

Rows 386-389: “It is beyond the scope of the present study to formulate a hypothesis about genetic penetrance of risk alleles of the HLA system, but it is noteworthy that all the risk alleles are evenly present in patients with T1D in our country, that have a lower percentage of European GA (supplementary table S5).”     This is very unclear. Should be rephrased.

Author Response

Response to Reviewer 2 Comments

RE: genes-2341716

Thank you for the opportunity to revise our manuscript titled “Association between HLA class II alleles/haplotypes and genomic ancestry in Brazilian patients with type 1 diabetes: a nationwide exploratory study”.

We have provided a point-by-point response to the reviewers’ comments and have amended our manuscript accordingly. Changes in the original version are tracked in the manuscript and marked in red in the point-by-point response below.

We hope that the revised manuscript addresses adequately all reviewers’ comments and is now suitable for publication in Genes. If the reviewers have any further comments, we would be happy to consider them. We look forward to hearing from you.

Yours sincerely,

Reviewer 2: The manuscript “Association between HLA class II alleles/haplotypes and genomic ancestry (GA) in Brazilian patients with type 1 diabetes: a nationwide exploratory study” by Gomes and co-workers describes a large nationwide study in Brazil where they aimed to identify HLA-DRB1, -DQA1 and -DQB1 alleles/haplotypes associated with European, African and Native American genomic ancestry in Brazilian patients with type 1 diabetes (T1D). They used a panel of 46 insertion/deletion markers around human genome developed to identify genomic ancestry of studied subjects. This analysis was able to measure the percentage of each genomic ancestry in individual genomes and revealed the strong admixture in the population. The study revealed that as a rule a higher percentage of European GA was observed in patients with risk alleles and haplotypes whereas African GA was higher in patients with protective alleles and haplotypes. The observed differences in risk and protective alleles and haplotypes are in concordance with earlier studies in different genetic backgrounds but the new approach able to genetically define the percentage of different genetic ancestries at individual level may be useful in estimation of T1D risk in admixed populations.

Response: We thank the reviewer’s comments for improving our manuscript.

Point 1: Introduction, page 3, row 68: …” the classes I alleles”…. Should be: the class I alleles.

Response 1: We have corrected the sentence.

Point 2: Rows 74 and 76: Citations 8 and 9 in the text are apparently changed in the reference list.

Response 2: We have revised it accordingly.

Point 3: Rows 85 and 86: …” take in account”….It should be “take into account”

Response 3: We have corrected the sentence.

Point 4: Page 4, row 98:…” Moreover, this latter event was…”.   This might be rephrased.

Response 4: We have rephrased the sentence as follows: “In that study, we found a low proportion of Native American ancestry, and, only 6.2% of the studied people presented ratios greater than 95% for one single ancestry. Moreover, the European GA was the only one that had a high proportion (>95%) in that sample.”

Point 5: Methods, HLA genotyping, page 6, row 179: …”PCR-RSSO”…. RSSO should be opened.

Response 5: We have modified it accordingly.

Point 6: Row 180:….CWID….. CWID should be opened. Common Well Documented

Response 6: We have modified it according to the reviewer’s recommendation. 

Point 7: Page 9, row 228…”In this sample”…. The word sample might be replaced by e.g. analysis.

Response 7: We have modified it according to the reviewer’s recommendation. 

Point 8: Discussion, page 15, rows 340-341: “The aforementioned data must be aligned with the actual propose for the rule of HLA system”,      This sentence should be clarified.

Response 8: Thanks for your comments. We have rephrased the sentence to clarify as follows “Our results are in agreement with Sanchez-Mazas group statement in the complementary information of the HLA system as a tool for anthropological studies [26]”.

Point 9: Rows 359-360: “In general, the risk of this allele is related to the presence of Caucasian alleles in the haplotype (DRB1:03:01 and DRB1*04)”…This is unclear. How can the risk of DRB1*09:01 be related to the presence of DRB1*03:01 and DRB1*04?

Response 9: Thanks for your comments. We have rephrased the sentence to clarify as follows: “In general, this allele conferred risk in the presence of Caucasian risk alleles in the genotypes, such as DRB1:03:01 and DRB1*04”.

Point 10: Page 16, row 372: “and the protective alleles”… There is one protective allele and then haplotypes. Should apparently be “protective alleles and haplotypes”

Response 10: We have modified it accordingly.

Point 11: Rows 386-389: “It is beyond the scope of the present study to formulate a hypothesis about genetic penetrance of risk alleles of the HLA system, but it is noteworthy that all the risk alleles are evenly present in patients with T1D in our country, that has a lower percentage of European GA (supplementary table S5).” This is very unclear. Should be rephrased.

Response 11We have rephrased the sentence as follows “It is beyond the scope of the present study to formulate a hypothesis about genetic penetrance of risk alleles of the HLA system in patients with T1D, but it is noteworthy that some patients carrying the risk alleles of the HLA system had a lower percentage of European GA (supplementary table S5).”